# A Retrospective Comparative Study of Endoscopic Treatment of Gastrocnemius Contracture using the Modified Soft Tissue Release Kit

**DOI:** 10.3390/medicina59030635

**Published:** 2023-03-22

**Authors:** Yiming Li, Zengguang Wang, Yaokai Gan, Xin Jiao, Chen Xu, Jie Zhao, Kerong Dai

**Affiliations:** 1Department of Orthopaedic Surgery, Shanghai Ninth People’s Hospital, Shanghai Jiao Tong University School of Medicine, Shanghai 200011, China; 2Shanghai Key Laboratory of Orthopaedic Implants, Shanghai Ninth People’s Hospital, Shanghai Jiao Tong University School of Medicine, Shanghai 200011, China

**Keywords:** gastrocnemius contracture, modified soft tissue release kit, minimal invasion, endoscopy, safety

## Abstract

*Background and Objectives*: This study aimed to evaluate the effectiveness and safety of endoscopic gastrocnemius recession using the self-developed Modified Soft Tissue Release Kit. *Materials and Methods:* This retrospective review followed up 22 patients (34 feet) who underwent endoscopic surgery and 20 patients (30 feet) who received open surgery between January 2020 and January 2022. The American Orthopedic Foot and Ankle Society (AOFAS) ankle-hindfoot score and the maximum ankle dorsiflexion angle were evaluated preoperatively and at the last follow-up. Postoperative complications were recorded. Patient satisfaction was surveyed at the last follow-up. The comparison between quantitative data was analyzed with the Wilcoxon signed-rank test. The comparison between qualitative data was analyzed with the chi-square test. *Results:* There was no significant difference in the baseline characteristics between the two groups. The AOFAS score in the endoscopic group increased from 50 (18) points preoperatively to 90 (13) points at the last follow-up; the maximum ankle dorsiflexion angle increased from −7.7 (2.8) degrees to 10.6 (3.6) degrees. The AOFAS score in the open group improved from 47 (15) points preoperatively to 90 (18) points at the last follow-up; the maximum ankle dorsiflexion angle increased from −7.6 (4.0) degrees to 10.7 (3.3) degrees. The change values of the AOFAS scores in the endoscopic and open groups were 39 (15) and 40.5 (11) points, respectively, and there was no significant difference between them. The change values of the maximum ankle dorsiflexion angles in the endoscopic and open groups were 19.5 (4.3) and 19.1 (4.9) degrees, respectively, and there was no significant difference between them. There were no complications, such as sural nerve injury, in both groups. There was no significant difference between the two groups in satisfaction with the surgical outcome. *Conclusions:* Endoscopic gastrocnemius recession using the Modified Soft Tissue Release Kit can significantly improve the foot function with significant mid-term efficacy and high safety.

## 1. Introduction

Gastrocnemius contracture is a common disease that causes ankle dysfunction. Approximately 65% of patients with non-neuropathic lesions of the midfoot and forefoot have gastrocnemius contracture [1]. The patients with gastrocnemius contracture mainly show limitations of ankle dorsiflexion. The severe cases even present with the equinus deformity, which seriously affects activities of daily life [1,2]. The Silfverskiöld test is often used to diagnose gastrocnemius contracture and distinguish it from Achilles tendon contracture. The following are the general criteria for the Silfverskiöld test. Soleus contracture is considered to be present if the maximum ankle dorsiflexion angle is less than 10° at both 90° knee flexion and full knee extension. This indicates the contracture of the entire Achilles tendon. Gastrocnemius contracture is considered to be the only condition if the maximum ankle dorsiflexion angle is greater than 10° at 90° knee flexion and less than 10° at full knee extension [3,4,5]. Gastrocnemius contracture is often divided into spastic and non-spastic types in the clinic. Spastic gastrocnemius contracture is caused by cerebral palsy, spinal cord injury, stroke, etc., and most patients have muscle lesions or denervation of muscle. Non-spastic gastrocnemius contracture is often associated with plantar fasciitis, flatfoot, hallux valgus, etc. [1,6,7,8]. For the treatment of non-spastic gastrocnemius contracture, the traditional open gastrocnemius recession is still the gold standard [9]. However, with the rapid development of endoscopic technology in recent years, minimally invasive treatment of gastrocnemius contracture with endoscopy is highly recommended due to its advantages of less trauma, shorter recovery time and less postoperative scarring [10,11,12]. Our department has applied endoscopic technology to the treatment of gastrocnemius contracture since 2018. In addition, we innovatively combined our self-developed Modified Soft Tissue Release Kit with endoscopic gastrocnemius recession, hoping to reduce surgical trauma and improve the safety of the endoscopic procedure while ensuring the effectiveness of the surgery. In this study, we compare endoscopic gastrocnemius recession using the Modified Soft Tissue Release Kit with open surgery to study the safety and effectiveness of the new technique, hoping to provide a new method for the treatment of gastrocnemius contracture.

## 2. Methods

This study was approved by the Ethics Committee of Shanghai Ninth People’s Hospital, Shanghai Jiao Tong University School of Medicine (approval number SH9H-2021-T444-1), and the privacy of patients was strictly protected during the study. In this study, we included patients who received endoscopic or open gastrocnemius recession in our department between January 2020 and January 2022. In the endoscopic group, 26 patients (42 feet) were included, and 22 patients (34 feet) were followed up. In the open group, we included 25 patients (38 feet) and followed up 20 patients (30 feet).

### 2.1. Sample Size and Study Design

This study used the difference test: α = 0.05, β = 0.1; the postoperative maximum ankle dorsiflexion angles in the open and endoscopic surgeries were 23.2 ± 6.9 and 14.7 ± 6.7, respectively, in previous literature [13,14]; PASS 15 (NCSS, US) software was used to calculate the sample size of each group, which was 15 cases. Therefore, the sample size of this study met the requirement.

This study was a retrospective, comparative study. The patients included in this study were divided into an endoscopic group and an open group according to the different surgical methods, and both groups contained underage patients. As no studies had shown that age was an influencing factor for the efficacy of gastrocnemius recession, and the sample size of this study was relatively small, we did not separate the underage and adult patients when analyzing the data.

### 2.2. Inclusion and Exclusion Criteria

Inclusion criteria: Patients diagnosed with gastrocnemius contracture with the Silfverskiöld test; patients who underwent endoscopic or open gastrocnemius recession in our department; the Strayer procedure was used for endoscopic or open gastrocnemius recession; all patients were followed up for at least 6 months.

Exclusion criteria: Patients with spastic gastrocnemius contracture caused by cerebral palsy, spinal cord injury, stroke, etc.; patients who received concurrent Achilles tendon lengthening; patients with pain, deformity or limited mobility in the lower extremity joints caused by osteoarthritis or gout; patients with a history of surgery or trauma to the lower extremity.

### 2.3. Modified Soft Tissue Release Kit

This kit was independently modified by us. It was composed of a thick (7 mm) cannula with a lining core, thin (6 mm) cannula with a lining core, spade knife, hook knife, detacher, thin (6 mm) expander and thick (7 mm) expander (Figure 1). This kit was originally designed for the endoscopic treatment of carpal tunnel syndrome. On the basis of the proven efficacy and safety of endoscopic release of the transverse carpal ligament using this kit, we decided to enlarge its applied area to the endoscopic release of the gastrocnemius aponeurosis.

### 2.4. Surgical Procedure

Endoscopic surgery: After successful anesthesia, the patient was placed in the supine position. The surgeon disinfected and draped the patient. The blood of the upper limb was evacuated using a dispersed blood bandage, and a tourniquet was used to prevent blood return. The line between the midpoint of the popliteal fossa and the posterior edge of the lateral malleolus was marked as the location of the sural nerve. According to the Strayer level, an approximate 0.8 cm transverse incision was made on the posterolateral calf at the level of 10–12 cm above the medial malleolus. The incision should avoid the sural nerve. After incising the skin and subcutaneous tissue, the soft tissue on the surface of the gastrocnemius aponeurosis was separated with the detacher, and attention was paid to carefully separating and protecting the sural nerve. Above the detacher, the expander was used to expand to the posteromedial calf in a direction perpendicular to the long axis of the gastrocnemius aponeurosis. The surgeon inserted the cannula and pulled out the lining core. The assistant held a hemostat to clip the balance blade on the cannula and made the long axis of the cannula perpendicular to the long axis of the gastrocnemius aponeurosis. Observation with a 2.7 mm, 30-degree endoscope showed white gastrocnemius aponeurosis above. The surgeon carefully confirmed that there was no nerve tissue in the operation area. Then, the dorsiflexion of the ankle was maintained by the assistant, and the surgeon completely incised the gastrocnemius aponeurosis with the hook knife or the spade knife under the direct endoscopic vision. After the incision, the underlying muscular tissue was exposed. The surgeon checked that the ankle dorsiflexion was significantly improved, indicating that the release of the gastrocnemius aponeurosis was satisfactory. After rinsing the incision, it was closed using 1–2 stitches with a 3-0 absorbable suture. (Figure 2).

Open surgery: The preparation for open and endoscopic surgery was the same before making the incision. However, the incision for the open surgery was approximately 3–4 cm, which was made on the posteromedial calf at the level of 10–12 cm above the medial malleolus according to the Strayer level. After incising the skin and subcutaneous tissue, the gastrocnemius aponeurosis was exposed with the retractor by the assistant. The soft tissue on the surface of the gastrocnemius aponeurosis was separated with the detacher, and attention was paid to carefully separating and protecting the sural nerve. In the direction perpendicular to the long axis of the gastrocnemius aponeurosis, the surgeon incised the gastrocnemius aponeurosis from medial to lateral with the scalpel or tissue scissors under direct vision. If the calf was thick or the exposure of deep tissue was difficult at this level, the surgeon could extend the incision or add an incision on the opposite side to facilitate the complete release of the gastrocnemius aponeurosis. The surgeon checked that the ankle dorsiflexion was significantly improved, indicating that the release of the gastrocnemius aponeurosis was satisfactory. After rinsing the incision, it was sutured with a 3-0 absorbable suture.

### 2.5. Outcome Measures

#### 2.5.1. American Orthopedic Foot and Ankle Society (AOFAS) Ankle-Hindfoot Score

The AOFAS score was selected as a subjective scale to assess the patients’ foot symptoms and function. The patients were evaluated preoperatively and at the last follow-up. All the patients in this study also had flatfeet, so this scale score was affected by the therapeutic outcome of the flatfoot. Therefore, the AOFAS score was used as a secondary indicator to evaluate the efficacy of the gastrocnemius recession.

#### 2.5.2. Maximum Ankle Dorsiflexion Angle

The maximum ankle dorsiflexion angle was the main indicator to evaluate the efficacy of the gastrocnemius recession. The patients were measured preoperatively and at the last follow-up. Specific measuring methods: With the help of the doctor, the ankle was placed in the maximum dorsiflexion while the knee remained extended. The two arms of the protractor were then placed along the fifth metatarsal and the long axis of the fibula, respectively. The maximum ankle dorsiflexion angle was 90° minus the measured angle. When the ankle was in the neutral position, the ankle dorsiflexion angle was 0. This angle was negative when the ankle was in plantar flexion. Similarly, this angle was positive when the ankle was in dorsiflexion.

#### 2.5.3. Patient Satisfaction

The satisfaction question was used on a five-point scale from 1 (very dissatisfied) to 5 (very satisfied). The patients were questioned at the last follow-up.

#### 2.5.4. Complications

Complications, such as poor wound healing, decreased gastrocnemius strength and sural nerve injury, were recorded.

### 2.6. Statistical Analysis

SPSS 26.0 software (IBM, Armonk, NY, USA) was used to perform the data analyses. The Shapiro–Wilk test was used for the normal distribution of the quantitative data. The quantitative data with a normal distribution were represented as the mean ± standard deviation. Similarly, the quantitative data with a non-normal distribution were represented as the median (interquartile range). The comparison between quantitative data was analyzed with the Wilcoxon signed-rank test, and *p* < 0.05 was considered statistically significant. The qualitative data were represented as the frequency and constituent ratio. The comparison between qualitative data was analyzed with the chi-square test, and *p* < 0.05 was considered statistically significant.

## 3. Results

### 3.1. General Results

A total of 42 patients (64 feet) were followed up. A total of 22 patients (34 feet) were treated with endoscopic gastrocnemius recession using the Modified Soft Tissue Release Kit, including 12 males (54.5%) and 10 females (45.5%). The mean age in the endoscopic group was 19.9 ± 12.6 years (range: 8–53 years), and the mean follow-up time was 19.1 ± 8.6 months (range: 7–32 months). A total of 20 patients (30 feet) were treated with open gastrocnemius recession, including 10 males (50%) and 10 females (50%). The mean age in the open group was 32.4 ± 22.8 years (range: 6–74 years), and the mean follow-up time was 13.9 ± 5.9 months (range: 7–26 months). In addition, all patients enrolled in this study were diagnosed with flatfoot combined with gastrocnemius contracture. Therefore, all patients received flatfoot correction in addition to the gastrocnemius recession. In the endoscopic group, 13 patients (24 feet) received subtalar arthroereisis, and nine patients (10 feet) received Evans lateral lengthening calcaneal osteotomy. In the open group, 10 patients (19 feet) received subtalar arthroereisis, and 10 patients (11 feet) received Evans lateral lengthening calcaneal osteotomy. The baseline characteristics of the two groups are detailed in Table 1.

### 3.2. Clinical Outcomes

A total of 22 patients (34 feet) were followed up in the endoscopic group. The median AOFAS score increased from 50 (18) points preoperatively to 90 (13) points at the last follow-up, and the difference was statistically significant (*p* < 0.001). The median maximum ankle dorsiflexion angle increased from −7.7 (2.8) degrees preoperatively to 10.6 (3.6) degrees at the last follow-up, and the difference was statistically significant (*p* < 0.001) (Table 2). A total of 20 patients (30 feet) were followed up in the open group. The median AOFAS score increased from 47 (15) points preoperatively to 90 (18) points at the last follow-up, and the difference was statistically significant (*p* < 0.001). The median maximum ankle dorsiflexion angle increased from −7.6 (4.0) degrees preoperatively to 10.7 (3.3) degrees at the last follow-up, and the difference was statistically significant (*p* < 0.001) (Table 3).

In the endoscopic group at the last follow-up, the AOFAS scores were rated as excellent in 24 feet (70.6%) and good in eight feet (23.5%). In the open group at the last follow-up, the AOFAS scores were rated as excellent in 19 feet (63.3%) and good in six feet (20.0%). The median change values of the AOFAS scores in the endoscopic and open groups were 39 (15) and 40.5 (11) points, respectively, and there was no significant difference between them (*p* = 0.322). The median change values of the maximum ankle dorsiflexion angle in the endoscopic and open groups were 19.5 (4.3) and 19.1 (4.9) degrees, respectively, and there was no significant difference between them (*p* = 0.856) (Table 4).

At the last follow-up, 20 (90.9%) and 17 (85.0%) patients in the endoscopic and open groups, respectively, were satisfied or very satisfied with the surgical efficacy. The satisfaction scores of the two groups are shown in Table 5. The statistical analysis showed that there was no significant difference between the endoscopic group and the open group in the satisfaction of the surgical efficacy (*p* = 0.497).

### 3.3. Complications

Postoperative wound healing was good in both groups without redness, swelling, exudation, etc. At the last follow-up, no patient reported plantar flexion weakness, and no patient indicated paresthesia or numbness in the sural innervation area.

## 4. Discussion

In this study, we expanded the applied area of our self-developed Modified Soft Tissue Release Kit to endoscopic treatment of gastrocnemius contracture. This new technique showed excellent clinical results with significant improvements in postoperative AOFAS scores and maximum ankle dorsiflexion angles. The new technique showed no significant difference in the change values of the two indicators compared with the gold standard. In addition, this new technique significantly improved the safety of endoscopic gastrocnemius recession, and there were no complications, such as sural nerve injury.

### 4.1. Levels of Gastrocnemius Recession

Conservative treatments, such as stretching exercises, could be performed in the early stage of gastrocnemius contracture. However, surgical treatment is required when conservative treatment is ineffective or functional limitation is severe [9]. In clinical practice, gastrocnemius recession is rarely performed alone and is usually performed in conjunction with other foot operations [14]. According to the different levels of recession, the methods of gastrocnemius recession mainly include the Abbassian procedure to release the proximal gastrocnemius, Baumann procedure to release the middle gastrocnemius and Strayer, Vulpius and Hoke procedures to release the distal gastrocnemius [15,16,17,18]. For the Abbassian procedure, only the medial head aponeurosis of the gastrocnemius was released, the incision was relatively small, and the sural nerve would not be injured during the operation. As a result, the correction of the maximum ankle dorsiflexion angle was usually not ideal after the Abbassian procedure, so it was rarely used [15]. For the Baumann procedure, its incision was located in the medial upper third of the calf. This procedure was mainly to release the muscle part of the gastrocnemius. The gastrocnemius could be first released in one place during the operation. If the surgeon was not satisfied with the results of ankle dorsiflexion, the surgeon continued to release the gastrocnemius in two or three places. The Baumann procedure was characterized by a relatively large incision and obvious postoperative scar, but the maximum ankle dorsiflexion angle was significantly corrected after surgery. In addition, there were few complications, such as weakened muscle strength of the lower limb, in this surgical method, so it was widely used [16]. As for the Vulpius procedure, its incision was located in the lower third of the soleus muscle belly. The gastrocnemius tendon was at this level. During the operation, the surgeon transversely incised the gastrocnemius tendon completely and released part of the muscle belly of the soleus. Although the maximum ankle dorsiflexion angle was obviously corrected after surgery, complications such as weakened plantar flexion strength were likely to occur. Therefore, this procedure was applicable to the plantar flexion deformity caused by the joint contracture of gastrocnemius and soleus, which was rarely used in patients with only gastrocnemius contracture [17]. For the Strayer procedure, its release level was located at approximately 2 cm distal to the most distal aspect of the gastrocnemius muscle belly. Only the gastrocnemius aponeurosis was incised during the operation, which did not affect the soleus strength at all. Therefore, the Strayer procedure, which guarantees the surgical efficacy while not causing decreased plantar flexion strength, coupled with the ease of the operation, has now become the most widely used procedure for gastrocnemius recession [18]. All patients enrolled in this study underwent the Strayer procedure to incise the gastrocnemius aponeurosis. The Hoke procedure was to release and lengthen the Achilles tendon, which was used primarily in patients with Achilles tendon contracture.

### 4.2. Current Status of Endoscopic Gastrocnemius Recession

Up to now, open gastrocnemius release is still the gold standard in the surgical treatment of gastrocnemius contracture. With the rapid development of endoscopic technology and the popularization of the minimally invasive concept in recent years, the use of endoscopic gastrocnemius recession has gradually increased. Phisitkul et al. used the dual-portal endoscopic gastrocnemius recession for minimally invasive treatment of gastrocnemius contracture. They followed up 320 patients treated with this technique for an average of 13 months. The maximum ankle dorsiflexion angle was increased from −0.8 ± 5.4 degrees preoperatively to 11.0 ± 6.6 degrees postoperatively. The VAS, SF-36 and FFI scores were significantly improved compared with those preoperatively. However, the incidence of postoperative complications, such as sensory disturbance or numbness in the sural innervation area and mild reduction in plantar flexion strength, was 6.6% (21/320) [14]. Saxena et al. treated 47 patients (54 feet) with single-portal endoscopic gastrocnemius recession. After a mean follow-up of 27 months, the maximum ankle dorsiflexion angle increased from −8 ± 4 degrees preoperatively to 7 ± 4 degrees postoperatively, but the incidence of sural nerve injury was 11.1% (6/54) [19]. Previous studies have even shown that the incidence of sural nerve injury in endoscopic gastrocnemius recession is as high as 13% to 17% [10,11,20] compared with less than 5% for open gastrocnemius recession [1,12,21,22].

### 4.3. Advantages of Endoscopic Gastrocnemius Recession Using the Modified Soft Tissue Release Kit

Therefore, although the efficacy of endoscopic gastrocnemius recession is exact, how to improve the safety of the endoscopic operation is the main problem to be urgently solved. Based on this, we applied the Modified Soft Tissue Release Kit, which had previously been used for endoscopic carpal tunnel release and had achieved great outcomes, to endoscopic gastrocnemius recession. This new technique has the following advantages: First, this kit is simple in design and easy to operate; second, an incision of only approximately 0.8 cm is made on the fibular side of the lower leg during the operation with less trauma and less obvious scarring; third, unlike other cannulas used in endoscopic surgery, we added a pair of balance blades to the cannula to prevent inadvertent rotation of the cannula intraoperatively, and during the operation, the balance blade was clipped with a hemostat by the assistant to ensure the stabilization of the cannula, making the operation safer; fourth, we also designed a deep sliding groove on the cannula to prevent the knife from accidentally leaving the operating area, which could reduce the risk of the operation; and fifth, there were two knives in this kit, a hook knife and a spade knife, and the surgeon could choose the knife more conducive to the operation according to his own habits or the intraoperative actual situation.

In this study, the patients who underwent endoscopic gastrocnemius recession with the Modified Soft Tissue Release Kit showed significant improvement in the maximum ankle dorsiflexion angle and the AOFAS score after at least six months follow-up. There was no significant difference between the endoscopic group and the open group in the change values of the maximum ankle dorsiflexion angle. Similarly, there was no significant difference in the change values of the AOFAS score between the two groups. Therefore, the effectiveness was great in endoscopic gastrocnemius recession using the Modified Soft Tissue Release Kit, and there was no significant difference compared with the gold standard operation. More critically, there were no complications, such as sural nerve injury, in this novel technique, which was evidently lower than previous studies. Hence, this new technique obviously improved the safety of the endoscopic gastrocnemius recession. Furthermore, compared with the gold standard operation, this new technique had the advantages of a smaller incision, less trauma and less evident scarring. Thus, this new technique is worth further popularization and application.

### 4.4. Experience in Applying the New Technique

Next, we would like to share some experience and advice on the clinical use of this new technique. First, the patient’s ankle was positioned in dorsiflexion with the help of the assistant to maintain the tension of the gastrocnemius aponeurosis during the procedure. Second, before inserting the cannula, the soft tissue on the surface of the gastrocnemius aponeurosis was separated with the detacher as far as possible. The cannula is then placed below the gastrocnemius aponeurosis along the upper side of the detacher. This can guarantee that the sural nerve was located below the cannula and outside the operating area, ensuring the safety of the operation. Third, attention should be paid to protecting the muscular tissue above the gastrocnemius aponeurosis to avoid excessive release during the operation.

### 4.5. Limitations of the Study

There were some limitations to this study. First, the sample size of this study was relatively small, so we did not discuss the results separately according to different ages or different flatfoot correction procedures. Second, this study was a retrospective study, which was conducted in a single center. Therefore, we plan to conduct a multicenter, randomized, controlled trial with a larger sample size in the future to obtain high-level data supporting the clinical efficacy of this new technique.

## 5. Conclusions

In conclusion, endoscopic gastrocnemius recession using the Modified Soft Tissue Release Kit has less trauma and a better appearance with a convenient operation and high safety. It can obviously improve foot and ankle function with significant middle-term efficacy. This new technique is expected to be further popularized and become a novel option for minimally invasive treatment of gastrocnemius contracture in the future.

## Figures and Tables

**Figure 1 medicina-59-00635-f001:**
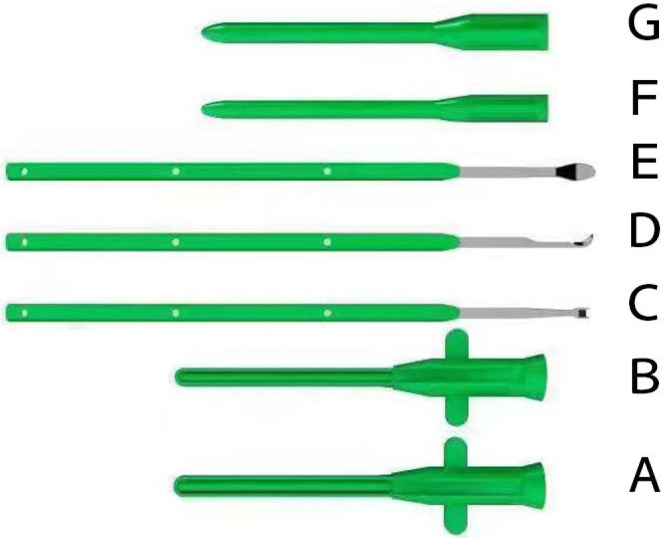
Modified Soft Tissue Release Kit. (**A**) Thick (7 mm) cannula with a lining core; (**B**) Thin (6 mm) cannula with a lining core; (**C**) Spade knife; (**D**) Hook knife; (**E**) Detacher; (**F**) Thin (6 mm) expander; (**G**) Thick (7 mm) expander.

**Figure 2 medicina-59-00635-f002:**
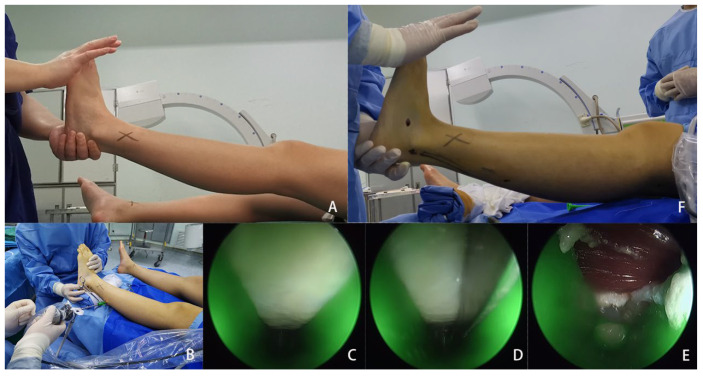
Surgical images of the endoscopic gastrocnemius recession using the Modified Soft Tissue Release Kit. (**A**) Preoperative examination showed obvious limitation of ankle dorsiflexion; (**B**) A 2.7 mm, 30° endoscope was used for observation; (**C**) The entire gastrocnemius aponeurosis was observed in the endoscopic view; (**D**) The gastrocnemius aponeurosis was incised with the hook knife; (**E**) The muscular tissue can be observed after the complete release of the gastrocnemius aponeurosis; (**F**) The ankle dorsiflexion angle improved significantly postoperatively.

**Table 1 medicina-59-00635-t001:** Baseline characteristics of the two groups.

Baseline Characteristics	Endoscopic Group	Open Group	*p* Value
Sex			0.768
Male	12 (54.5%)	10 (50%)	
Female	10 (45.5%)	10 (50%)	
Age (years)	19.9 ± 12.6 (8–53)	32.4 ± 22.8 (6–74)	0.117
Follow-up time (month)	19.1 ± 8.6 (7–32)	13.9 ± 5.9 (7–26)	0.056
Foot			0.638
Left	15 (44.1%)	15 (50%)	
Right	19 (55.9%)	15 (50%)	
Flatfoot correction procedure			0.537
Subtalar arthroereisis	24 (70.6%)	19 (63.3%)	
Evans osteotomy	10 (29.4%)	11 (36.7%)	

**Table 2 medicina-59-00635-t002:** The outcomes of AOFAS scores and maximum ankle dorsiflexion angles in the endoscopic group.

Indicators	Preoperative	Last Follow-Up	*p* Value
AOFAS score (points)	50 (18)	90 (13)	<0.001
Maximum ankle dorsiflexion angle (degrees)	−7.7 (2.8)	10.6 (3.6)	<0.001

Values are presented as the median (interquartile range). AOFAS score: American Orthopedic Foot and Ankle Society ankle-hindfoot score.

**Table 3 medicina-59-00635-t003:** The outcomes of AOFAS scores and maximum ankle dorsiflexion angles in the open group.

Indicators	Preoperative	Last Follow-Up	*p* Value
AOFAS score (points)	47 (15)	90 (18)	<0.001
Maximum ankle dorsiflexion angle (degrees)	−7.6 (4.0)	10.7 (3.3)	<0.001

Values are presented as the median (interquartile range). AOFAS score: American Orthopedic Foot and Ankle Society ankle-hindfoot score.

**Table 4 medicina-59-00635-t004:** The change values of the AOFAS score and maximum ankle dorsiflexion angle in the two groups.

Indicators	Endoscopic Group	Open Group	*p* Value
Change values of the AOFAS score (points)	39 (15)	40.5 (11)	0.322
Change values of the maximum ankle dorsiflexion angle (degrees)	19.5 (4.3)	19.1 (4.9)	0.856

Values are presented as the median (interquartile range). AOFAS score: American Orthopedic Foot and Ankle Society ankle-hindfoot score.

**Table 5 medicina-59-00635-t005:** Satisfaction with the surgical outcomes of the two groups at the last follow-up.

Levels	Endoscopic Group	Open Group	*p* Value
Very satisfied—5-point	18	13	0.497
Satisfied—4-point	2	4	
Fair—3-point	2	2	
Dissatisfied—2-point	0	1	
Very dissatisfied—1-point	0	0	

## Data Availability

The data presented in this study are available on request from the corresponding author. The data are not publicly available due to privacy reasons.

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
