# Peer review of "A Retrospective Comparative Study of Endoscopic Treatment of Gastrocnemius Contracture using the Modified Soft Tissue Release Kit"

_medicina, 2023, doi:10.3390/medicina59030635_

Round 1

Reviewer 1 Report

I was very pleased to read the manuscript entitled "A Retrospective Comparative Study of Endoscopic Treatment of Gastrocnemius Contracture using the Modified Soft Tissue Release Kit". It is a well-written article with a suitable methodology. 

I have a few doubts about the data presentation, which should be explained.

1) In both the introduction and the discussion part authors claim, that "Endoscopic gastrocnemius recession using the Modified Soft Tissue Release Kit 32 can significantly improve the foot function, with significant mid-term efficacy and high safety." In the results, there are no significant differences between open and endoscopic procedures, and therefore, this claim should be reformulated, to be less misleading.

2) Have you got data about the average surgery time for both the open and endoscopic method groups?

Reviewer 2 Report

Dear Authors,

thanks for the submitted paper, I found it interesting for the topic and, almost all, well structured.

However, I found some critical issues that I list below.

1.     you compared two patient groups, respectively 22 patients (34 feet) versus 20 patients (30 feet): did you perform a power analysis to support these numbers? please discuss it.

2.     Age range was respectively 6-74 years and 8-53 years. Do you think pediatric age can invalidate the general result? Did you plan to include only adult patients or consider children and adolescents separately? Do you think this could constitute a bias? Please argue

3.     in the introduction paragraph, you correctly described that gastrocnemius contracture can be related to neurological pathologies or other ankle or foot deformities. In performing procedures as you describe, was the gastrocnemius procedure always performed as an isolated procedure or in combination with other surgical times, involving the foot or ankle? Please discuss.

4.     Limitation section is not exhaustive, even considering the points suggested by my revision report.

Since the topic is interesting and the manuscript can help many colleagues, I recommend a global review of methods section (patient selection, inclusion/exclusion criteria, statistical analysis…), which can improve the quality of the manuscript.

In its current form, in my opinion the paper does not deserve publication in this valuable journal. 

I hope, however, that Authors will perform a comprehensive review and subsequent re-submission in a form which may be suitable for publication.

Round 2

Reviewer 1 Report

Agree. Thanks for your reply.

Reviewer 2 Report

Dear Authors,

you answered in the cover letter file, but I think in an incomplete and unsatisfactory way.

The answers to my questions were not included in the text, making no changes which, in my opinion, would have increased the value of the paper.

1.     you compared two patient groups, respectively 22 patients (34 feet) versus 20 patients (30 feet): did you perform a power analysis to support these numbers? please discuss it.

You should add to the manuscript

2.     Age range was respectively 6-74 years and 8-53 years. Do you think pediatric age can invalidate the general result? Did you plan to include only adult patients or consider children and adolescents separately? Do you think this could constitute a bias? Please argue, also you should discuss in the manuscript

3.     in the introduction paragraph, you correctly described that gastrocnemius contracture can be related to neurological pathologies or other ankle or foot deformities. In performing procedures as you describe, was the gastrocnemius procedure always performed as an isolated procedure or in combination with other surgical times, involving the foot or ankle? Please discuss. 

Nowhere in the manuscript is it stated that each patient also underwent flatfoot correction. This section is lacking. You should also report the type of treatment performed, mentioning, at a minimum, the procedures performed. In my opinion this aspect is important and should be reported in the text.

4.     Limitation section is not exhaustive, even considering the points suggested by my revision report. 

The limitations paragraph has not been implemented as suggested in the previous revision.

Since the topic is interesting and the manuscript can help many colleagues, I recommend a global review of methods section (patient selection, inclusion/exclusion criteria, statistical analysis…), which can improve the quality of the manuscript.

Having re-read the new version of the manuscript, I do not note the comprehensive revision I had suggested, since several aspects were not revised.

At current state, I confirm, in my opinion the paper does not deserve publication in this valuable journal. 

I strongly believe that it is more important to give priority to submitting an accurate version, rather than a quick one, also out of respect for the Reviewers and Editors, who dedicate their time and knowledge to reading the manuscripts.

Round 3

Reviewer 2 Report

I appreciate your efforts to edit and improve the manuscript, favorably; it deserves, in my opinion, publication in this esteemed Journal